

# Calibration of a semi-distributed lumped karst system model and analysis of its sensitivity to climate conditions: the example of the Qachqouch karst spring (Lebanon)

Emmanuel Dubois[1], Joanna Doummar[2*], Séverin Pistre[3], Marie Larocque[1],

5 [1]Université du Québec à Montréal, Département des sciences de la Terre et de l'atmosphère, Pavillon Président-Kennedy, local PK-6151 C.P. 8888, Succursale Centre-Ville Montréal (Québec) H3C 3P8 Canada
[2]Department of Geology, American University of Beirut, PO Box 11‑0236/26, Beirut, Lebanon
[3]HSM, Univ. Montpellier, CNRS, IRD, 300, avenue du Professeur Emile Jeanbrau 34000 Montpellier, France

*Correspondence to: Joanna Doummar (jd31@aub.edu.lb)

10 **Abstract.** Flow in complex karst aquifers is challenging to conceptualize, therefore to model for better management practices, especially in poorly investigated areas, in semi-arid climates, and under changing climatic conditions. The objective of this work is to propose a calibration approach based on time-series analyses for a karst aquifer and to assess the impact of changing climate conditions on the spring discharge. Based on more than three years of high-resolution continuous monitoring, a semi-distributed lumped model was calibrated and validated for the Qachqouch karst spring, north of Beirut (Lebanon). Time-series analyses and decomposition of spring hydrographs revealed that the system has a high regulatory function, with considerable storage capacity providing stable flow (minimum flow of 0.2 m3/s) during the dry season, and with flow rates exceeding 10 m3/s during the wet season, similar to other karst aquifers in the region. Based on this detailed understanding of the hydrodynamics of the system, the model geometry and parameters were validated. Three linear reservoirs were implemented to reproduce the combined contribution of the different flow components of the system. A satisfactory simulation (Nash-Sutcliffe coefficient = 0.72) of measured spring flow rates was obtained after calibration. Climate change conditions (+1 to +3°C warming, -10 to -30% less precipitation annually, and intensification of rain events) were added to a baseline climatic year to produce scenarios of expected spring flow responses. Results show that the Qachqouch karst aquifer is sensitive to decreasing rainfall, which is associated with more pronounced recessions, with flow rates decreasing by 34% and 1-month longer dry periods. Because of the limited influence of snow on the spring flow rate, a warming climate has less impact on spring flow conditions than a reduction in precipitation. Although the model shows that increasing rainfall intensity induces larger floods, recessions and shorter low flow periods, the real impact of high-intensity precipitation events remains uncertain, since the model does not account for complex unsaturated and epikarst processes. This work shows that calibrating a semi-distributed lumped model using time series analysis can be an efficient approach to improve simulations of complex karst aquifers, thus providing useful models for long-term sustainable water management.





## 1 Introduction

Around the world, karstic aquifers are strategic water resources that have been used to provide water to populations since early civilizations (Chen et al., 2017a; Ford and Williams, 2007). This is particularly true in Mediterranean areas (Bakalowicz, 2015), where water resources are scarce. The exploitation of karst aquifers remains a challenge because of their highly heterogeneous nature (Bakalowicz, 2005; Ford and Williams, 2007; Stevanović, 2015) and their high vulnerability to contamination from anthropogenic activities, especially under global change conditions (Chen et al., 2018; Doummar and Aoun, 2018a, b; Hartmann et al., 2014; Iván and Mádl-Szőnyi, 2017; Leduc et al., 2017). A decrease in precipitation rates and an increase in temperature have already been experienced in the Mediterranean Basin for the last four decades (Milano et al., 2013), and are expected to worsen according to the climate change scenarios established for the 21st century (Collins et al., 2013; Giorgi and Lionello, 2008; Kirtman et al., 2013).

Hydrogeological modelling of karst systems is a useful approach for sustainable aquifer management, as it allows spring discharge to be simulated (Fleury et al., 2009; Li et al., 2016) and provides a means to account for the impacts of climate change on future flow rates (Chen et al., 2018; Hartmann et al., 2012, 2014). Among the different types of available models, lumped models simplify karst systems (epikarst, conduit network, porous or fissured matrix, superposed karstic series) into a schematic representation of interconnected reservoirs to reproduce spring discharge (Ghasemizadeh et al., 2012; Hartmann et al., 2013; Mazzilli et al., 2017; Sauter et al., 2006). Since these models do not usually require spatial variations in karst hydrodynamic properties and subsurface processes (Chen and Goldscheider, 2014), they are best suited for poorly-characterized karst hydrosystems. These models usually require meteorological data and sometimes river flow data as input (Bailly-Comte et al., 2008, 2010, 2012, Larocque et al., 1998). Usually, their calibration is based on the manual or automatic adjustment of empirical parameters that represent connections between the reservoirs, to reproduce measured spring discharges to the extent possible using rainfall time series as input (Chen and Goldscheider, 2014; Doummar et al., 2012). However, these models are often calibrated to reproduce only the observed outputs of the systems, without accounting for subsurface discretization and distributed physical flow processes (Kong-A-Siou et al., 2013).

In the last 40 years, time-series analyses of karst aquifers have been widely used to enhance the understanding of karst aquifer dynamics (Bailly-Comte et al., 2008; Fiorillo, 2014; Jeannin and Sauter, 1998; Larocque et al., 1998; Mangin, 1984), providing information on karst aquifer storage, groundwater travel times, and hydrogeological properties. By including these inferred processes, models of karst aquifers have been successfully calibrated to reproduce karst aquifer groundwater levels, spring flow rates, and chemographs (Adinehvand et al., 2017; Chen and Goldscheider, 2014; Hartmann et al., 2013; Hosseini et al., 2017; Larocque et al., 2000; Schmidt et al., 2014), thereby enhancing their representativeness and usefulness. However, the utility of lumped models calibrated to reproduce only spring flow rates can be limited to estimate water vulnerability to contamination, especially in karstified areas, because input data are not spatially distributed and the flow processes are highly simplified.





The objectives of this work were 1) to illustrate how a semi-distributed lumped model can be calibrated using a robust knowledge of the hydrodynamic functioning of a karst aquifer derived from statistical and correlation time-series analyses and 2) to estimate its sensitivity to climate conditions. The approach is demonstrated on the Qachqouch karst spring in the

region north of Beirut (Lebanon), which is governed by semi-arid conditions. In 2014, a high-resolution monitoring network was established in the Qachqouch spring catchment area, which previously had only been poorly studied (Doummar and Aoun, 2018a, b; Dubois, 2017). A semi-distributed lumped model developed using MIKE SHE (DHI, 2016a, b) is calibrated here using observed spring discharge time series. Fitting parameters were inferred and refined based on hydrodynamic properties derived from autocorrelation and cross-correlation analyses. The calibrated model was used to assess the

sensitivity of flow rates in this karst system to potential variations related to climate change.

## 2 Field Site

The Qachqouch spring, located in the Metn area in Lebanon, 18 km north of Beirut, drains a catchment of approximately 56 km2, with a northern boundary (shared with the Jeita spring catchment) delineated by the Nahr el-Kalb River (Fig. 1). The topography of the upstream catchment area is relatively mountainous, with elevation ranging between 60 to more than 1,500

meters above sea level (m asl). The catchment of the Nahr el-Kalb River was determined topographically (Margane and Stoeckl, 2013), while the hydrogeology of the Qachqouch spring was further investigated by Dubois (2017). A minor hydrogeological connection was established between the Nahr el-Kalb River and the Qachqouch spring from repeated tracer experiments and micropollutant analyses (Doummar and Aoun, 2018a, b).

Similar to the nearby Jeita spring (Margane et al., 2013, 2018), the Qachqouch spring originates from the Jurassic karst

aquifer at about 64 m asl (Fig. 1). The Jurassic formation is mainly comprised of a 1,070 m-thick massive limestone sequence with intertonguing dolostones in the lower parts of the formation resulting from diagenetic dolomitization (Hahne, 2011; Margane et al., 2013; Nader et al., 2007). The catchment area is under the influence of the Yammouneh fault regime (El Hakim, 2005, Hahne, 2011), leading to a high degree of tectonic deformation and fracturing. Sea level variations, especially the Messinian salinity crisis, and quaternary glaciations also contributed to creating a high level of karstification in

the Mediterranean area in several stages (Bakalowicz, 2015; Nehme et al., 2016) and the Jurassic limestone (Margane et al., 2013) with karst features, such as lapies, sinkholes, drains, and caves.

The spring is highly polluted due to excessive non-sorted solid waste and untreated wastewater disposal upstream in its urbanized catchment (Doummar and Aoun, 2018a, b). However, during low flow and drought periods, the spring is used to supplement domestic water use in Beirut and its surrounding areas. The spring outlet consists of a 150 m-long concrete

tunnel, thereby offering a suitable cross-section for flow measurement, except during extensive floods where access to the spring is inundated.

The average yearly precipitation is estimated to be 1,000 mm from one station deployed over the Qachqouch catchment at 950 m asl (local high-resolution monitoring ongoing since 2014), while the average yearly temperature is 16.2°C.



Precipitation occurs mostly between November to April, and snow is observed in January and February (above 1,000 m asl),
with only limited snow accumulation over long periods in the Qachqouch catchment (Dubois, 2017). Rainfall events are usually intensive, with considerable amounts occurring over relatively short periods (e.g., 461 mm in January 2019, 97 mm on December 24, 2017, and 43 mm in one hour on October 30, 2017). The average daily air temperature varies between 5°C during winter and 23°C during summer with occasional sub-zero temperatures.

## 3 Materials and methods

### 3.1 General approach

The conceptualization of the Qachqouch system (Fig. 2) was developed based on the collection of high-resolution data for over three years and on the subsurface characterization of the system (box 1, Fig. 2). Flow rate classification was performed to qualitatively characterize the system's functioning (box 2, Fig. 2), and time series decomposition, simple and cross-correlation analyses were performed to estimate the system memory and response time (box 3, Fig. 2). Using this
information, a semi-distributed linear reservoir model of the Qachqouch system was conceptualized and parameterized (box 4, Fig. 2). Climate change scenarios were elaborated (box 5, Fig. 2) based on Intergovernmental Panel on Climate Change (IPCC) projections for 2040 to test possible future system responses under different scenarios based on the calibrated and validated model (box 6, Fig. 2).

### 3.2 Meteorological and hydrological data

In the framework of a monitoring project funded by USAID since 2014 (PEER, Cycle 3), two full climatic stations (one *Campbell-Scientific-Alpine* station with a heated gauge and one *Hobo* station mounted with a data logger) were installed at elevations of 1,700 and 950 m asl respectively (hourly precipitation, temperature, humidity, wind direction and speed, and solar radiation) in the region of the Qachqouch and Jeita springs. Additionally, a multi-parameter probe (*In situ-Troll 9500*) was installed at the spring discharge to record water level and water temperature every 30 minutes. The discharge flow rates
were calculated using a rating curve based on bimonthly measured discharge rates by direct gauging with errors of 8-10% during recession periods. Flow rates exceeding 10 m$^3$/s are considered to be less precise, especially when the section is overflowed. The few gaps in the time series (less than one month combined) were linearly interpolated and smoothed using a moving average over the entire time series to remove outliers and aberrant values.

For the purposes of this study, temperature (-0.45°C/100 m elevation) and precipitation (6%/100 m elevation) gradients were
calculated through a regression analysis of the incremental and cumulative variations in temperature and precipitation with altitude between the two climatic stations (Doummar et al., 2018).

### 3.3 Time series analyses

### 3.3.1 Analyses of the spring flow rate data



A frequency analysis was used to characterize spring flow rates (Dörfliger et al., 2010; Mangin, 1971; Marsaud, 1997). A
log-normal distribution was used to link a given flow rate to its frequency of measurement, except for outliers arising from
variation in flow dynamics.

Following hydrograph decomposition, the method developed by Mangin (1971, 1975) to estimate the dynamic volume
($V_{dyn}$) available in the aquifer during the depletion flow of a karst spring was used. As shown in Eq. (1) to (3), this method
separates the recession of the hydrograph ($\Psi$) from depletion ($\varphi$). It is the equivalent of considering two reservoirs, the first
associated with the vadose zone, which drains into the second reservoir, corresponding to the phreatic zone. The total spring
flow rate (Q) is divided into:

$$Q(t) = \Psi(t) + \varphi(t) \tag{1}$$

The two parts are detailed in:

$$\Psi(t) = q_0 \cdot \frac{1 - \eta \cdot t}{1 - \varepsilon \cdot t} \tag{2}$$

$$\varphi(t) = Q_{R0} \cdot e^{-\alpha t} \tag{3} \text{ (known as Maillet equation)}$$

Finally, the dynamic volume is obtained by:

$$V_{dyn} = c \cdot \frac{Q_i}{\alpha} \tag{4}$$

with $q_o$ being the infiltrating flow at the beginning of the flood event [$L^3.T^{-1}$]; $\eta$ the infiltration velocity coefficient [$T^{-1}$]; $\varepsilon$ the
heterogeneity flow coefficient [$T^{-1}$]; $t$ the time from the beginning of the flood event [T]; $Q_{Ro}$ the fictional flow rate of the
depleting curve at the maximum of the flood event [$L^3.T^{-1}$]; $\alpha$ the depleting coefficient [$T^{-1}$]; $Q_i$ the flow rate [$L^3.T^{-1}$] at time
$t_i$ when depletion began ($\eta = 1/t_i$); and $c$ the time constant (86,400 s/d for $Q_i$ in m$^3$/s and $\alpha$ in d$^{-1}$)

The calculation of the parameters listed above allows the two parameters, $k$ and $i$, used in the karst spring classification
proposed by Mangin (1975) to be characterized for comparison with other Middle Eastern and Lebanese springs analyzed by
El-Hakim and Bakalowicz (2007) and Bakalowicz et al. (2008). The parameter $k$ (Eq. (5)) characterizes the extent of the
phreatic zone and its regulating capacity; its processes of storage and discharge of infiltrated rainfall. Karstic springs are
supposed to have $k < 0.5$ (Mangin, 1975; Marsaud, 1997), as opposed to $k > 0.5$ characteristic of porous aquifers.

$$k = {V_{dyn}} \Big/ {V_{trans}} \tag{5}$$

with $V_{dyn}$ being the dynamic volume [$L^3.T^{-1}$] and $V_{trans}$ the average annual transit volume [$L^3.T^{-1}$].

A system with mainly fast infiltration would be characterized by an $i$ close to 0, while a system with mainly slow and
delayed infiltration would have an $i$ value of close to 1. The parameter $i$ is estimated using the following ratio at $t = 2$ days
(Eq. (6)):



$$i = \frac{1 - \eta.(t)}{1 + \varepsilon.(t)} \tag{6}$$

### 3.3.2 Decomposition of the spring hydrograph

Jeannin and Sauter (1998) suggested, based on Forkasiewicz and Paloc (1967), to consider three reservoirs in
conceptualizing flow from a fractured rock aquifer: a reservoir with low permeability (reservoir 1), a conduit network
(reservoir 2), and an intermediate system (reservoir 3). Each of these can be represented using the exponential drainage of a
reservoir based on Darcy's law, with specific recession coefficients (Eq. (7)).

$$Q(t) = Q_1.e^{-\alpha_1.t} + Q_2.e^{-\alpha_2.t} + Q_3.e^{-\alpha_3.t} \tag{7}$$

with $Q_1$, $Q_2$, and $Q_3$ being the initial discharge rates of each reservoir [$L^3.T^{-1}$], and $\alpha_1$, $\alpha_2$, and $\alpha_3$ the reservoir recession
coefficients [$T^{-1}$], linked to their respective hydraulic conductivities and geometries (Kovács et al., 2005; Fiorillo, 2011).

### 3.3.3 Correlation analyses

The autocorrelation function of a time series $r(p)$ is obtained by measuring the linear dependency of a given signal with itself
(Eq. (8)), shifted by the time lag ($p$), and is deduced from the correlogram $C(p)$ (Eq. (9)).

$$r(p) = \frac{C(p)}{C(0)} \tag{8}$$

$$C(p) = \frac{1}{n} \sum_{t=1}^{n-p}(x_t - \bar{x})(x_{t+p} - \bar{x}) \tag{9}$$

with $C(0)$ the correlogram for $p = 0$, $p$ the time lag (p = 0 to m); $n$ the length of the time series; $x$ a single event; $\bar{x}$ the mean
of all events; $m$ the cut-off point, which determines the interval over which the analysis is carried out, usually chosen to
circumscribe a given hydrograph feature, such as annual or long-term effects.

Mangin (1984) suggested that the analysis is only valid until m = n/3. Based on the autocorrelation function, the memory
effect represents the inertia of the system and the possibility of storage. It is identified as the time when $r(p)$ reaches a small
value and is used to compare karstic hydrosystems relative to one another. Here, this value is set to 0.2, as suggested by
Mangin (1984).

The correlation between an input signal, $x(t)$, and an output signal, $y(t)$, is given by a cross-correlation function, $r_{xy}(p)$ (Eq.
(10)), obtained from the cross-correlogram $C_{xy}(p)$ (Eq. (11)).

$$r_{xy}(p) = \frac{C_{xy}(p)}{\sigma_x \sigma_y} \tag{10}$$

$$C_{xy}(p) = \frac{1}{n}.\sum_{t=1}^{n-p}(x_t - \bar{x})(y_{t+p} - \bar{y}) \tag{11}$$





with $\sigma_x$ and $\sigma_y$ being the standard deviations of the time series. The delay of a system is the lag between p = 0 and the peak of cross-correlation between input and output.

### 3.3.4 Numerical modelling of a semi-distributed linear reservoir

The Qachqouch system was modelled using the MIKE SHE software (DHI 2016a; 2016b). The modelled catchment is subdivided spatially into several sub-catchments, in addition to the saturated zone, which is represented as several linked reservoirs (Fig. 3). The model is decomposed into three domains (Doummar et al., 2012). The first domain consists of the atmosphere, composed of rainfall and snow time series spatialized over the spring catchment and its topography (Fig. 3). The second domain entails a spatialized simplified unsaturated zone (UZ), consisting of land use, vegetation, and crop
coefficients, such as leaf area index (LAI) and root density (RD) to calculate potential evapotranspiration (PET) based on Penman-Monteith (FAO, 1998), water interception by vegetation (Kristensen and Jensen, 1975), and runoff. The second domain also includes the unsaturated zone, represented using a simple two-layer model to calculate actual evapotranspiration (AET) and subsequent infiltration through the unsaturated zone. The volume of water infiltrating into the third domain, which is the saturated zone (SZ), is then simulated at each time step. The SZ is simulated via linear reservoirs comprised of
five interflowing reservoirs, corresponding to the five major sub-catchments of the spring (I1 to I5; Fig. 3), which empty into each other (in topographical order), and includes two base flow reservoirs (B1 and B2; Fig. 3). In this configuration, discharge from the entire system feeding the Qachqouch spring consists of the cumulative discharge of spatially discretized connected reservoirs feeding into a fast flow and a slow flow reservoir with different recession coefficients. The complete model is therefore considered a classical lumped model for the saturated zone, but also a physically-based model for the
system inputs (atmosphere) and the soil compartment.

The model uses 20 parameters defining the atmosphere, the UZ, and the saturated interflow and base flow reservoirs. The simulated flow rates were compared to measured values using the Nash-Sutcliffe coefficient (NS) and the root mean square error (RMSE) to achieve the best calibration possible with a calibration period (2015/01 – 2018/04) and a validation period (2018/05 –2019/07). Sensitivity analysis was conducted automatically on single parameters using the Autocal function (DHI,
2016) to identify the parameters to which the model is highly sensitive.

### 3.3.5 Sensitivity to future climatic variations

According to the IPCC (IPCC, 2014), at the 2080-2100 horizon, temperature in the eastern Mediterranean Basin will have increased from +2 to +3°C (Representative Concentration Pathway (RCP) 2.6 – optimistic scenario) to +5 to +7 °C (RCP 8.5 – pessimistic scenario) during the December-May rain season relative to the 20[th] century trend (Collins et al., 2013; Kirtman
et al., 2013). Here, the assessment of potential climate change impacts on spring flow rates is undertaken by applying several simple climate change conditions onto climate data from the observation period (average daily temperature and precipitation estimated from the mean observed values) for an average year for the catchment to simulate the expected spring discharge under future conditions. According to the global climate model (GCM) IPSL-CM5 and the mid-range RCP 6.0 scenario





(Dufresne et al., 2013) adapted for the Lebanese context (Doummar et al., 2018), the expected warming is +0.1 to +0.3°C per

year (*i.e.*, a total increase of +1 to +3°C above the current average annual temperature for the 10[th] year of the 2020-2030 period). A reduction in annual precipitation in the range of -10 to -30 % (Fig. 9) is expected with an increase in evapotranspiration proportional to the rise of temperature.

## 4 Results

### 4.1 Classification of spring flows: qualitative functioning of the system

**4.1.1 Measured flow rates**

The interannual (2015 to 2019) average Qachqouch spring flow rate is 2.1 $m^3$/s (1.7 $m^3$/s during calibration period and 3.4 $m^3$/s during validation), and its total annual discharge varies between 44 and > 50 $Mm^3$. The average flow rate is 2 $m^3$/s during high flow periods (during a normal year) and recedes to 0.2 $m^3$/s during recession periods, with a maximum recorded > 20 $m^3$/s for a short period of time following a flood event (there is a high degree of uncertainty as to the exact value of the

maximum flow rate since it is higher than 10 $m^3$/s). Flow rates for the 2018-2019 hydrological year were particularly high, corresponding to an exceptionally rainy year (1,800 mm between September 2018 and July 2019 in comparison to the average of 1,000 mm/year). Therefore, this period is only used as a modeling validation period and flow measurements are not included in the time series analysis.

The flow rate frequencies (Fig. 4) can be divided into four different categories based on the slopes of the frequency curve.

The interpretation suggested by Dörfliger et al. (2010), based on the shapes of the curves and the positions of the three breaking slope points, can be used to understand the aquifer hydrodynamics. The kink in slope between $a_3$ and $a_4$ shows that the section at the gauging station is probably overflowed for flow rates above 10 $m^3$/s. The flow rates corresponding to slopes $a_2$ (between 0.25 and 3 $m^3$/s) and $a_3$ (between 3 and 10 $m^3$/s) are interpreted as the presence of two distinct temporary storages within the aquifer. Flow rates of less than 0.25 $m^3$/s (break in slope between $a_1$ and $a_2$) correspond to water that is

slowly released from the aquifer. This generally confirms that the flow rates are related to a single aquifer with two interacting base flow reservoirs corresponding to a capacitive and conductive function, or slow and fast flows, with no inter-aquifer exchanges, therefore facilitating the further interpretation of the time series analyses.

### 4.1.2 Recession coefficients

The coefficients $k$ and $i$, representing the regulation capacity of the system (storage/discharge of rainfall) and the infiltration

conditions (fast versus delayed) respectively, were calculated for the Qachqouch spring based on the 2015, 2016, and 2017 recessions (Fig. 5). Although the Qachqouch system has a high regulation capacity ($k$ = 4 years), similar to a porous aquifer, it has an average infiltration delay, $i$, of 0.5, characterizing a system with significant storage recharged by both fast and slow





infiltration, which classifies Qachqouch system among the other Lebanese and Middle Eastern karstic systems (El-Hakim and Bakalowicz, 2007).

**4.1.3 Decomposition of the spring discharge time series**

Assuming that the spring discharge can be represented using several draining reservoirs (with the drainage function following Darcy's law), 77 depletion coefficients (Mangin, 1975) were calculated over the entire time series (Fig. 6). These coefficients can be classified into three categories. In the first, the summer dry period has low flow rates, around 0.25 m$^3$/s, and slow depletion, characterized by $\alpha$ values of 0.004 d$^{-1}$ on average ($\alpha_1$). In the second category, flow rates are less than 7 m$^3$/s, while having faster depletion and $\alpha$ values of 0.050 d$^{-1}$ on average ($\alpha_2$). The highest flow rates are found in the third category, and occur at the beginning of the depletion, with very fast depletion related to $\alpha$ values of 0.220 d$^{-1}$ on average ($\alpha_3$).

**4.1.4 Correlation analysis**

The autocorrelation function of rainfall (performed with a maximum lag of 1/3 of the available time series) shows a rapid decrease to zero, while that of the spring flow rates shows a slow decrease, with a memory effect of approximately 50 days (Fig. 7a). This means that the system has a substantial regulation capacity, similar to that reported for other Lebanese springs (Bakalowicz et al., 2008; El-Hajj, 2008). The peak of the cross-correlation function of rainfall (input) against spring flow rates (output; also performed with a maximum lag of 1/3 of the available time series) is 0.52 (Fig. 7b). It is delayed by approximately one day, characterizing a fast infiltration component with a strong correlation to the rainfall events. The inertia of the rest of the system tends to smoothen the input signal, as portrayed by the slow decrease in the cross-correlation function from $r_{xy}(p) = 0.25$. An annual cycle (not fully displayed here) is visible with an increase in $r_{xy}(p)$ starting again at 270 days.

**4.2 Simulated past flow rates**

The discharge of the Qachqouch spring for the 2015 to 2019 hydrogeological years was successfully simulated using the integrated semi-distributed linear reservoir model (Fig. 8). The model was simulated since 2012 to allow for a spin up period before the calibration and validation time span (2015-2019). The calibrated model parameters are presented in Table 1. A satisfactory fit is observed between the simulated and the measured flow rates, with an NS of 0.70 (0.60 during calibration; 0.74 during validation) and a RMSE of 1.57 m$^3$/s (1.27 m$^3$/s during calibration; 2.17 m$^3$/s during validation). The average simulated flow rate is 2.0 m$^3$/s (1.7 m$^3$/s during calibration; 3.0 m$^3$/s during validation), very close to the average measured flow rate of 2.1 m$^3$/s (1.7 m$^3$/s during calibration; 3.4 m$^3$/s during validation), although the calibration period did not include the exceptional year. Even though the annual recession periods are well simulated by the model, the simulated summer flow rates reach values of lower than 0.1 m$^3$/s (for approximately 70 days); values that were not reached in the measured flow rates. The exceptional 2018-2019 year with high precipitation rates and an important spring flow response induces a relatively higher mean residual error. The parameters to which the model was found to be most sensitive (Table 1) are the



base flow reservoirs specific yields and time constant, the fraction of percolation to the base flow reservoir (reservoir B2
from Fig. 3), and the time constant for percolation between the interflow reservoirs. The previous analyses performed on the
time series allowed the model geometry (number of reservoirs and their parameters) and the ranges of these parameters to be
refined during calibration, thereby reducing model uncertainty by optimizing the conceptual model on the system
hydrodynamic functioning (Enemark et al., 2019).

Larocque et al. (2000) have shown that verifying the consistency of simulated flow rates in a karst aquifer is possible by
performing autocorrelation and cross-correlation analyses on the simulated values and comparing the results with the time
response functions of the observed data. The autocorrelation function of the measured and simulated flow rates are similar
here, with a simulated memory effect that is slightly longer (62 days) than that of the real system (50 days). The cross-
correlogram of the simulated spring discharge generally fits those of the measured discharges, with similar delays
(approximately one day) between input (rainfall) and output (spring discharge), and a similar shape of the cross-correlation
function (Fig. 7b). The model therefore generally reproduces both the fast and slow response of the system to infiltration
well.

## 4.3 Simulated future responses to climate changes

The climate change conditions derived from seven potential scenarios for the study area (Table 2) have been applied over ten
consecutive years to compare the system responses in the tenth year to a baseline scenario, obtained by averaging the
monitored climatic data (precipitation and temperature) on a daily basis to produce an average year (Fig. 9). The seven
climatic scenarios were calculated from the baseline by applying the respective climatic conditions to it.

Scenarios 1 and 2, with less precipitation, lead to notably lower annual discharge flows than the baseline scenario (11 to 34
% lower; Fig. 9 and Table 2). The reductions in spring discharge for the warming scenarios (scenarios 3 and 4) are not
pronounced (not represented on Figure 9). However, combining warmer temperatures with a reduction in precipitation
(scenarios 5 and 7) slightly increases the reduction in spring flow rates (12 to 36 % lower than for the baseline; Fig. 9 and
Table 2). The scenario with intensified precipitation events (scenario 7) produces higher peak flows than the baseline
scenario.

Although the duration of summer low flows (*i.e.*, the number of days where discharge is < 0.2 m$^3$/s) increases accordingly
with reductions in precipitation and rising temperatures, only scenarios 2 and 6, corresponding to a 30 % reduction in
precipitation, led to a notable increase in summer low flow duration, by more than a week per year (Table 2). Scenario 7,
with intensified precipitation events, led to a shorter duration of summer low flows than for the baseline scenario, as well as
to a slightly higher mean annual flow rate.





## 5 Discussion

### 5.1 Dynamic functioning of the spring: a comparative approach

Many authors have linked spring discharge from a karstic aquifer to the functioning and geometry of the system (e.g., Fiorillo, 2014) using various methods of spring recession analysis. In this study, the flow rate return probability provided qualitative information regarding the system functioning. This method shows that this karst system stores large volumes of water during high flow periods and releases water during the low flow period. This has been further investigated and confirmed through the decomposition of the spring hydrograph and the recession analyses that allowed fast and delayed

infiltration components to be identified. The latter was also observed in the autocorrelation and cross-correlation functions, with a slow decrease in the correlations over time and a short delay between the input and output signals (*i.e.*, cross-correlation peak).

These results provide a conceptual representation of the transfer function (Bailly-Comte et al., 2008; Marsaud, 1997). The classification of the spring hydrograph recessions into three categories (Fig. 4b) is interpreted as corresponding to three

major reservoirs representing different hydraulic conductivity zones (Fiorillo, 2011; Kovács et al., 2005) or two different hydraulic stages of the system (Fiorillo, 2014), including a transient stage as previously modelled by Kovács and Perrochet (2008) for synthetic karst spring hydrographs. It can be interpreted that after a rainfall event, a first reservoir or zone with high hydraulic conductivity evacuates water rapidly and produces high flow rates at the Qachqouch spring via a well-developed karstic drainage network (peak of the cross-correlation rainfall-spring discharge; Fig. 7b). Then, depending on the

season, a second reservoir or zone with lower hydraulic conductivity contributes to an intermediate depletion rate. Independently from the flow periods, the slowest transfer function continuously feeds the spring base flow, an influx of water that is associated with a third reservoir or zone, with the lowest hydraulic conductivity. As a result, these depletion coefficients are useful to describe the spring discharge and can be used as calibration parameters to which the model output is highly sensitive.

Even though the Qachqouch karst system has been reported to be less complex than that of the neighboring Jeita spring (Doummar, 2012; Margane et al., 2018), based on the karst system classification (parameters $k$ and $i$, representing the extent of the phreatic zone, its regulating capacity, and the type of infiltration – Bakalowicz et al., 2008; El-Hakim and Bakalowicz, 2007; Mangin, 1975), it is comparable to other Middle Eastern karst systems. In this region, karst aquifers are characterized by their relatively thick unsaturated zone and saturated aquifer (the sum of the respective thicknesses being greater than

1,000 m), leading to a high regulating capacity with a complex drainage structure (El Hakim, 2005; El-Hakim and Bakalowicz, 2007; El-Hajj, 2008; Hosseini et al., 2017; Petalas et al., 2018; Schmidt et al., 2014). The high regulation of the spring discharge transforms the input (mostly rainfall with occasional snowmelt) into annual to pluri-annual cycles and therefore constrains the cross-correlation function between the input and the output of the system to low values compared to other systems or similar values for systems in similar hydrogeological context (Table 3). This can be explained by the duality





of the stable low flow during the dry season (resulting from the redistribution of rainfall events to sustain continuous flow) and the flood flow following precipitation events (high flow rates during the wet season).

**5.2 Conceptual model of the Qachqouch system**

Although the model adequately reproduces flow discharge, it underestimates the summer low flows. Measurements recorded during flooding of the spring gauging station might be underestimated due to errors in the discharge water level rating curve

for high flow rates. Another explanation could be that the fast flow linked to a highly-developed drainage system is oversimplified in this reservoir model. The thickness of the UZ, combined with its lithological heterogeneity, may contribute to the relatively stable summer low flow by allowing considerable water storage. In fact, the dolostone could be compared to low-permeability porous media drained by a high-permeability system, thus allowing a large storage capacity in the upper parts of the aquifer. Furthermore, the high degree of karstification of the area, resulting from both the eustatic variations

(Messinien) and the quaternary glaciations, leads to a complex drainage system, with a probable paleo-network under the current base level. This would enhance the storage capacity of the system, as well as induce rapid flow rate increases (Bakalowicz, 2015; Nehme et al., 2016).

Based on water chemistry and stable isotopes, a link between the intermittent Nahr el-Kalb River and the Qachqouch system has been demonstrated (Doummar and Aoun, 2018a, b), but is not taken into account in this study. If the simulated flow rates

at the outlet are well represented, the link between the river and the aquifer is indirectly represented in the model, since the river could be considered to be an indirect source to the karst system, which depends on the same input as the model (precipitation). The river-aquifer link is probably determining for transport through the system.

**5.3 Dynamic functioning and spring responses to potential future climatic variations**

According to the simulated climate change scenarios, the climate context (semi-arid) and similarly to previous studies (Chen

et al., 2018; Hartman et al., 2012; 2015, 2017), a decrease in annual rainfall (-10% to -30%) could lead to lower discharge at the outlet, especially if combined with an increase in temperature (a 36% reduction in discharged volume), meaning that total precipitation drives spring discharge. The depletion time of the karst aquifer is expected to decrease, leading to summer water shortages (flow rates of less than 0.2 $m^3$/s during the dry season) occurring earlier in the season and lasting for longer periods of time (up to +25 days annually). Given the uncertainty on summer low flows (simulated past flow rates < 0.1 $m^3$/s

for the summer period, lower than any observed data at the outlet), the prediction of whether or not the spring would be dry for a period of time could not be ascertained with this model.

Although increasing temperatures leads to higher AET rates (Table 2), variations in temperature do not seem to significantly impact the modelled results, unlike Chen et al. (2018) for some of the studied springs. This can be explained by the fact that AET rates are already close to zero during the dry season (no rain between May to early November). It is also coherent with

the fact that spring flow is not substantially affected by snow melt, as catchment delineation and stable isotope data have shown (Doummar and Aoun, 2018b), unlike other major neighboring springs (Doummar et al., 2018).



The delay in the recession period and the fewer water shortage days simulated under the intensified precipitation scenario (scenario 7) are due to the distribution of more intense precipitation events on a daily time step that leaves the simulated soil reservoir frequently dry, even during the wet season, and therefore restricting the soil water consumption for AET. During the wet days, AET meets PET and the rest of the water goes mainly to recharge the system. Overall, as AET slightly decreases, recharge to the system slightly increases. It is important to emphasize that the model response to a high-intensity precipitation event would most likely change if simulations were run on an hourly time step. This is especially important when considering exceptionally intense precipitation events, creating flooding at the spring, such as the recorded 42.5 mm in one hour on October 30, 2017, which was followed by an unusual flood at the spring for this time of the year. More work is required to include climate change scenarios with a focus on changes in rainfall distribution patterns, and especially to include higher rainfall intensities during the flood season and less annual rainfall. However, the potential effect of rainfall intensification could be balanced out or exacerbated by the long-term evolution in catchment land use (e.g., effects of vegetation cover drying earlier in the season, reforestation programs, intensive landscaping, etc.). Climate change scenarios at the catchment scale should be coupled with land use evolution scenarios in the model to better understand the future functioning of the karst system.

## 6 Conclusion

The objectives of this work were to optimize the calibration of a karstic spring model, based on knowledge of the hydrodynamic functioning of the system, and to understand its sensitivity to climate conditions, with the example of the Qachqouch karst spring in Lebanon. Flow rates were analyzed to better conceptualize the system, and a semi-distributed linear reservoir model, calibrated based on this conceptualization, was used to reproduce current conditions and to analyze possible future climate conditions.

The results show that the system has significant storage capacity, enabling water flow to be sustained throughout the year. Although the karst aquifer has this inter-annual regulatory capacity, similarly to that of other Middle Eastern karst systems, it also has a substantial fast flow component due to its favourable geological context for karstification. The combination of these conditions explains the high contrast between high flow rates observed during the wet season ($>10$ m$^3$/s) and depletion reaching 0.20 m$^3$/s during the dry months.

Data analyses allowed a conceptual model to be developed through the decomposition of the time series into three reservoirs, which helped to calibrate the model. This comprehensive approach allowed the regular spring recession rates and durations and the input/output correlation to be satisfactorily reproduced, even for a year with exceptional precipitation rates, even though the model has been calibrated on a short period (3 hydrological years). Based on this model, the simulation of future climate conditions has shown that the spring flow rates are mainly sensitive to rainfall, with a non-linear response, with 36% losses in spring flows after 10 years of a 30% reduction in rainfall and summer low flow periods that could be extended by one month. The low sensitivity to increased temperatures is an indication that the Qachqouch spring may be more resilient to

warmer conditions than it is to dryer conditions. The impact of intensified precipitation events remains uncertain. The next
step should entail local coupling of climate change scenarios (at the catchment scale) with land use evolution scenarios to
improve overall future predictions, as the latter were not considered here.

This work shows that calibrating a model based on time series analyses and decomposition of the spring hydrograph is an
efficient methodology to enhance the quality of simulations for complex highly-karstified and poorly-investigated systems.
This approach could be applied to other karstic systems to simulate current or future flows, and is a compromise between
characterization efforts and simulation results when climate change impacts have not yet been assessed.

## 7 Author contribution

All authors contributed to developing the approach and writing the paper. Simulations were performed by JD and ED; figure
preparation by ED. Based on JD's funding acquisition, data was collected by JD's research team, which included ED at the
time. SP took on the publication fees.

## 8 Competing interests

No competing interests.

## 9 Acknowledgements

This work was funded by a USAID PEER Science project (award number 102881; Cycle 3) and the publication fees covered
by HydroScience Montpellier. The authors would also like to thank Fouad Andari and Michel Aoun for their help during
fieldwork. Mr. Roland Abourahal is thanked for providing space for the meteorological station and for facilitating local
contacts. The kind support of Engineer George El Kadi, Mr. Antoine Zoghbi, and the Mount Lebanon Water Establishment
is highly appreciated.

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

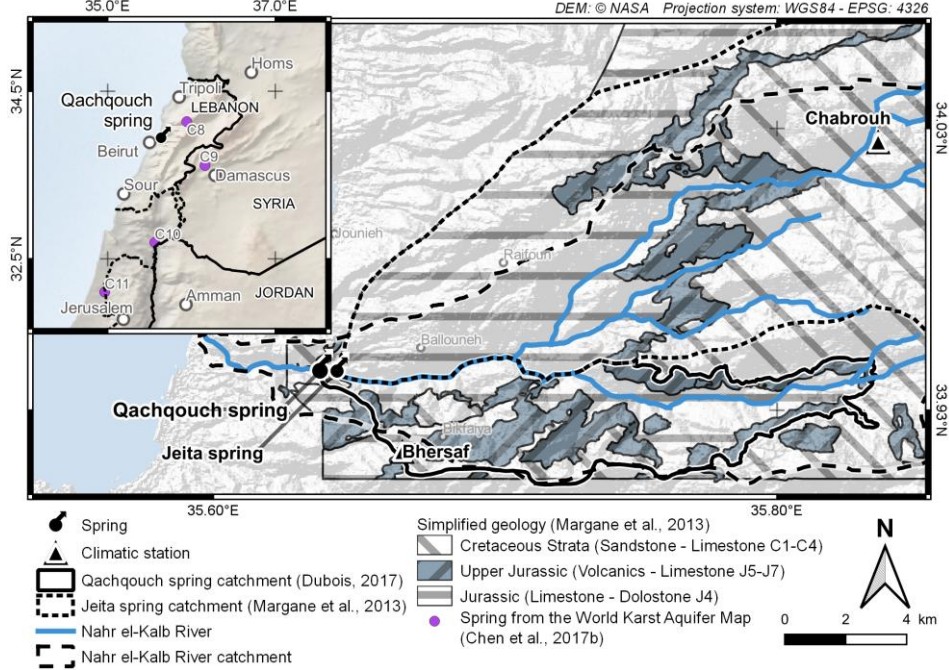

**Figure 1: Location of the Qachqouch karst spring near Beirut (Lebanon).**



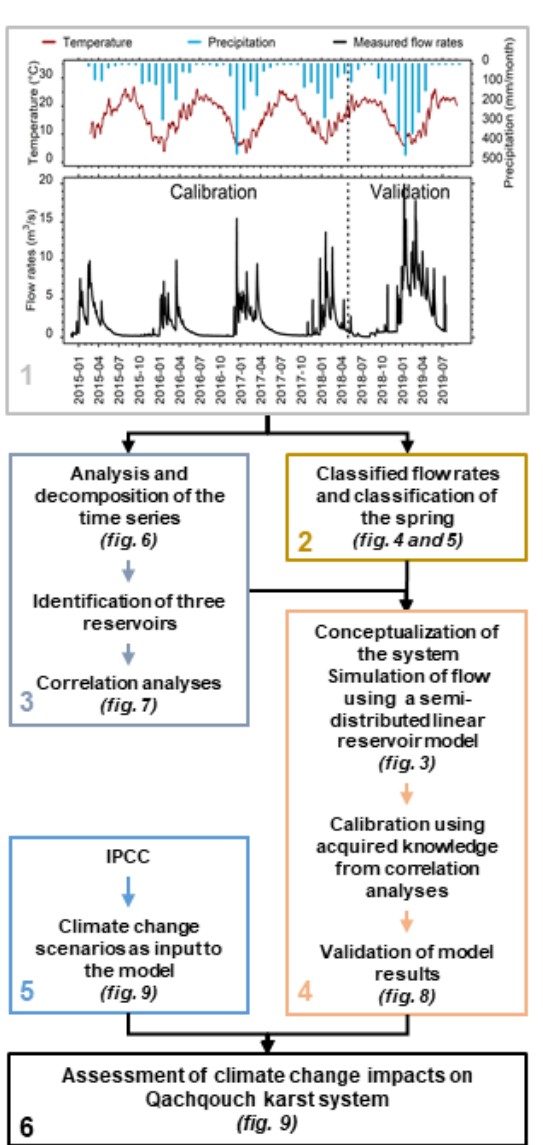

**Figure 2: Schematic representation of the six-step methodology adopted for model conceptualization and the calibration.**





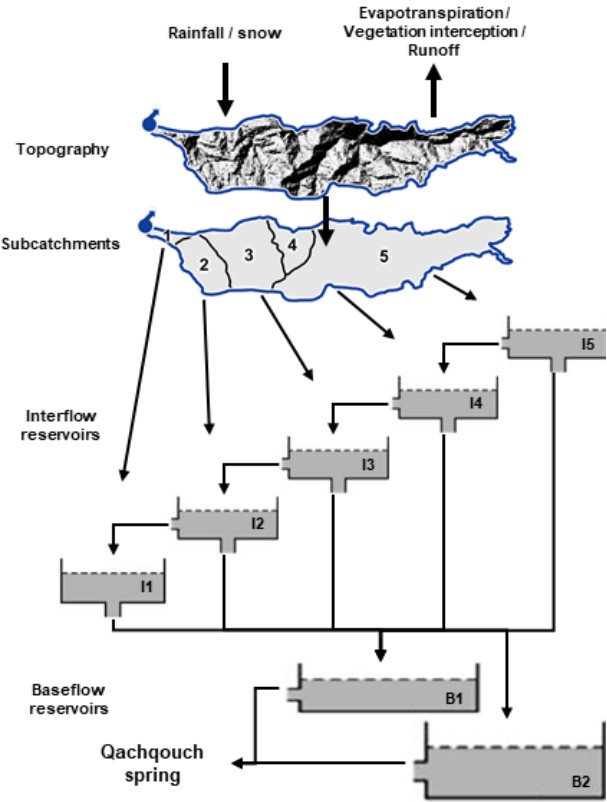

**Figure 3: Conceptual map of the model, using the internal structure of MIKE SHE (DHI, 2016a, 2016b).**

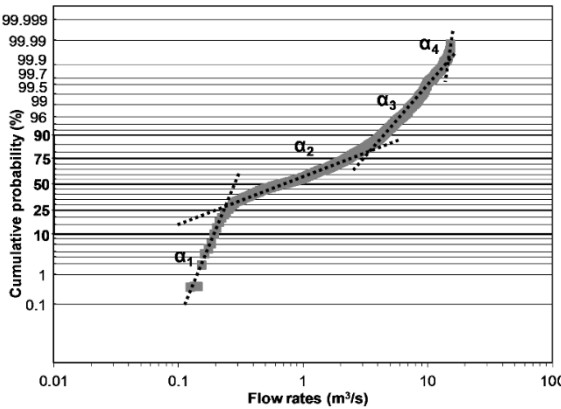

**Figure 4: Cumulative probability of flow rates and their classification by slope ($\alpha_1$ to $\alpha_4$) for the 2015-2018 period.**


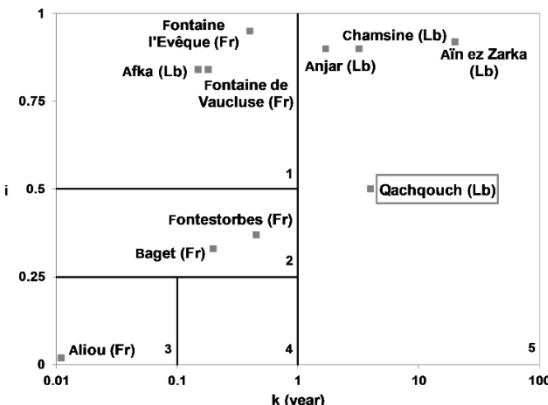

Figure 5: Qachqouch spring within the classification of karstic springs as a function of their k and i parameters (El-Hakim and Bakalowicz, 2007). Fr = France and Lb = Lebanon.

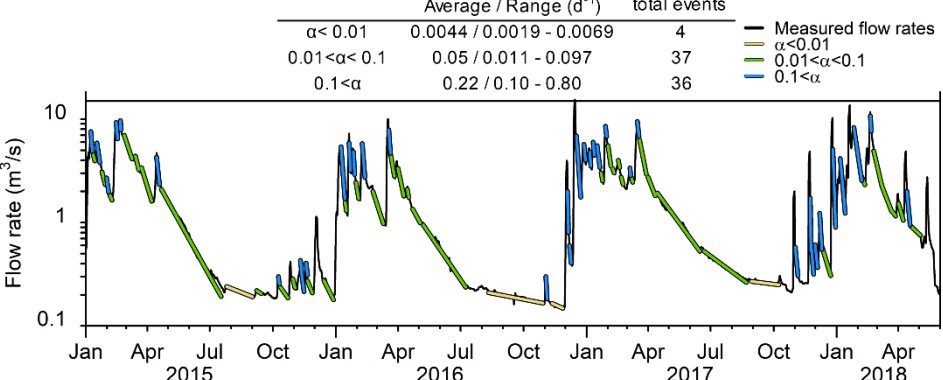

Figure 6: Calibration of the recession coefficients for 77 depletions over the 2015-2018 period.

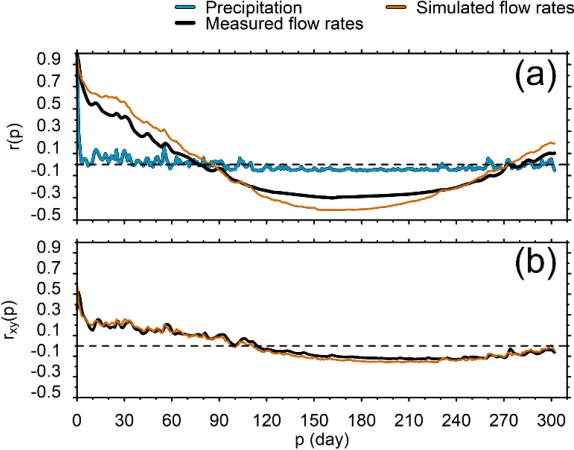

Figure 7: Autocorrelation functions of precipitation, measured flow rates, and simulated flow rates (a) and cross-correlation functions between precipitation (input) and measured or simulated flow rates (outputs) (b). Both were performed on 1/3 of the 2015-2018 period (m = n/3).





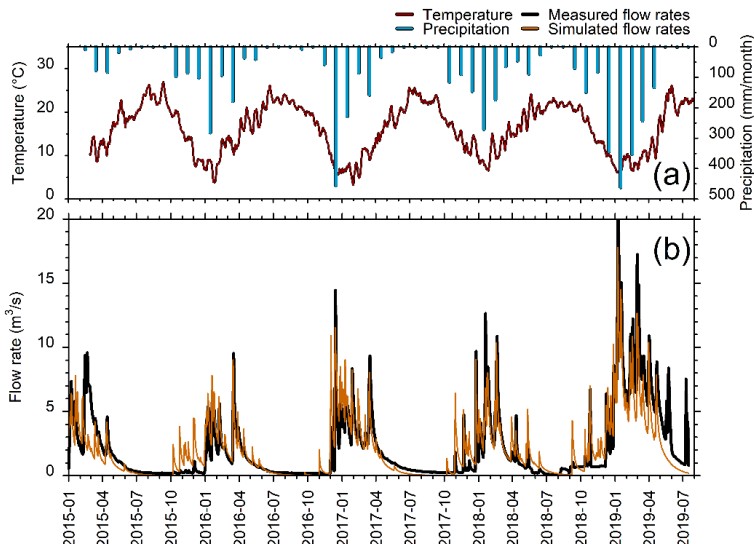

**Figure 8: Measured meteorological data (a) and observed and simulated spring discharge (b).**

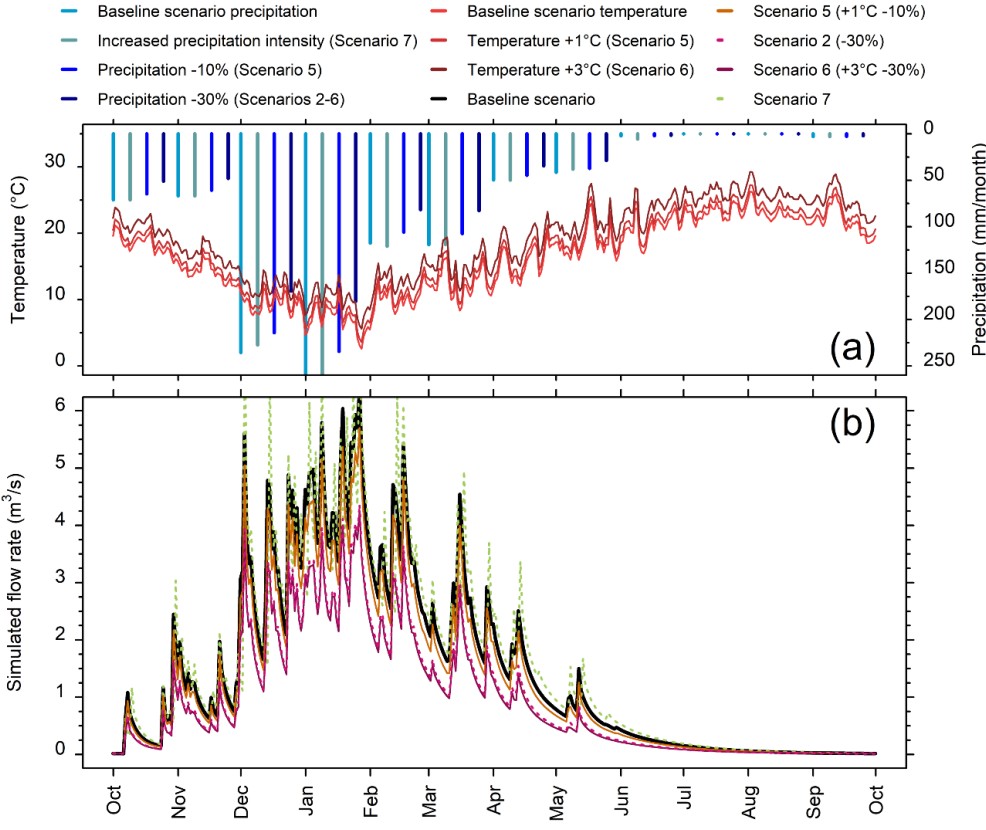


**Figure 9: Precipitation and temperature scenarios (a) and simulated flow rates (scenarios 1, 2, 5, 6, 7, and the baseline scenario) for the spring for the 10th hydrological year of the period 2020-2030 (b).**





**Table 1: Parameterization of the hydrological model (MIKE SHE 2016), final calibration values, and sensitivity tests.**

| Component | | Parameter | Value | Unit | Type of Data | Normalized sensitivity [-] | Ranking of parameters according to sensitivity analysis |
|---|---|---|---|---|---|---|---|
| **Atmosphere and Surface** | **Climatic data** | Precipitation (P) | | | Measured | <0.01 | 9 |
| | | Temperature (T) | - | mm | Measured | <0.01 | 9 |
| | | Potential Evapotranspiration (PET) | | | Calculated | <0.01 | 9 |
| | | Temperature lapse rate | -0.45 | °C/100m | Measured | <0.01 | 10 |
| | | Precipitation lapse rate | 6 | %/100m | Measured | -0.02 | 9 |
| | **Snow melt** | Melting temperature | 1 | °C | Measured | <0.01 | 10 |
| | | Melting rate | 2 | mm/°C /day | Calibrated | <0.01 | 10 |
| | **Vegetation** | Leaf Area Index (LAI) | 1 | [-] | Calibrated | 0.01 | 9 |
| | | Root Depth- (RD) | 5 | m | Calibrated | -0.08 | 7 |
| **Unsaturated Zone (UZ)** | **Soil** | Hydraulic Conductivity at saturation ($K_s$) | $1.10^{-6}$ | m/s | Calibrated | <0.01 | 10 |
| | | Field capacity | 0.2 | [-] | Calibrated | <0.01 | 10 |
| | | Total porosity | 0.3 | [-] | Calibrated | -0.19 | 5 |
| | | Wilting point | 0.05 | [-] | Calibrated | <0.01 | 10 |
| **Saturated Zone (SZ)** | **Interflow reservoirs (1 to 5)** | Specific yield | 0.01-0.1 | [-] | Calibrated | 0.03 | 8 |
| | | Time constant for percolation | 2-5 | day | Calibrated | -0.14-0.39 | 4 |
| | | Time constant (1/α) | 5 | day | Calibrated[1] | 0.04-0.11 | 6 |
| | **Base-flow reservoir 1** | Fraction of percolation to reservoir 1 | 0.3 | [-] | Calibrated | 0.23 | 4 |
| | | Specific yield | 0.2 | [-] | Calibrated | 0.83 | 2 |
| | | Time constant (1/α) | 20 | day | Calibrated[1] | 0.1 | 6 |
| | **Base-flow reservoir 2** | Specific yield | 0.15 | [-] | Calibrated | 1 | 1 |
| | | Time constant (1/α) | 200 | day | Calibrated[1] | 0.64 | 3 |

[1] - Previously calibrated with the analyses of the time series

**Table 2: Simulation results and comparison with the baseline scenario.**

| # | Name | Scenarios Condition | Precipitation mm/year | Average annual discharge flow mm/year | m³/s | in comparison with the baseline scenario | Actual evapotranspiration mm/year | Duration of the summer low flow (days < 0.2 m³/s) |
|---|---|---|---|---|---|---|---|---|
| - | Baseline | Invariant precipitation, temperature, and ET rates (average climatic year) | 966 | 837 | 1.49 | - | 167 | 112 |
| 1 | Precipitation -10 % | Cumulative decrease of precipitation rate over 10 years of 10% | 873 | 745 | 1.33 | -11 % | 163 | 118 |
| 2 | Precipitation -30 % | Cumulative decrease of precipitation rate over 10 years of 30% | 679 | 556 | 0.99 | -34% | 152 | 133 |
| 3 | Temperature +1°C | Cumulative increase of temperature over 10 years of 1°C and associated evolution of ET rates | 966 | 828 | 1.47 | -1% | 176 | 114 |
| 4 | Temperature +3°C | Cumulative increase of temperature over 10 years of 3°C and associated evolution of ET rates | 966 | 811 | 1.44 | -3% | 194 | 115 |
| 5 | Precipitation -10 % / temperature +1°C | Combination of 1 and 3 | 873 | 736 | 1.31 | -12% | 171 | 119 |
| 6 | Precipitation -30 % / temperature +3°C | Combination of 2 and 4 | 679 | 532 | 0.95 | -36% | 175 | 137 |
| 7 | Increased intensity of precipitation | Sum of the baseline precipitation per 3 days (annual precipitation constant) | 960 | 879 | 1.57 | +5% | 118 | 102 |

**Table 3: Maximum cross correlation ($r_{xy}(p)$) values between input and output signals and memory effects of karstic systems reported in the literature.**

| Reference | System | $r_{xy}(p)_{max}$ | memory effect (day) |
|---|---|---|---|
| This study | Qachqouch karst spring (Lebanon) | 0.52 | 50 |
| Marsaud, 1997 | Baget (France) | 0.6 | 15 |
| Larocque et al., 1998 | La Rochefoucauld karst aquifer (France) | 0.8 | 80 |
| El Hakim, 2005 | Anjar, Chamsine (Lebanon) | 0.3; 0.35 | 69; 90 |
| Bailly-Comte et al., 2008 | Coulazou river (France) | 0.8 | <20 |
| El-Hajj, 2008 | Dalleh, Bziza (Lebanon) | 0.45; 0.4 | 50; 40 |
| Hosseini et al., 2017 | Sasan aquifer (Iran) | 0.2 | 90 |
| Li et al., 2017 | Guzhou catchment area (China) | 0.45 | 3.5 |
| Petalas et al., 2018 | Paradisos Karst aquifer system (Greece) | 0.25 | 125 |