# Peer review of "Calibration of a lumped karst system model and application to the Qachqouch karst spring (Lebanon) under climate change conditions"

_Hydrology and Earth System Sciences, 2020_

## Referee Comment (RC1) · Nico Goldscheider (Referee) · 8 Apr 2020

General evaluation and recommendation:

The paper describes the calibration of a semi-distributed model for the simulation of the discharge of a large spring in Lebanon and uses the model to assess future impacts of climate change on groundwater resources. This is a relevant and timely topic. The paper is well prepared in any respect and suitable for publication in HESS following moderate revisions.

Specific comments:

Title: The title is too long (almost three lines). Please shorten to two lines.

Abstract: Something is wrong with the first sentence, which is also too long. Please rephrase.

16, 17 and elsewhere: m3 should be $m^3$

Introduction: Well written in general. Short, but all relevant aspects are included.

62-65: The research objectives are relevant, but maybe you could add 1-2 additional objectives. Objective no. 2 is very general and could be complemented by a more specific research question, also including the practical relevance of your research, such as the expected climate change impacts and the implications for freshwater availability. Furthermore, objective no. 2 is not completely clear. What do you mean by "its sensitivity" – the sensitivity of the model or of the karst aquifer?

73: km2 should be $km^2$

80: Rearrange sentence to avoid misunderstanding. The spring is located at 64 m asl, not the aquifer.

84: quaternary should be Quaternary

84: The expression "high level of karstification" is misleading in this case. In fact, the Messinian salinity crisis created a very low topographic level of karstification. You probably mean high degree of karstification, very intense and very deep karstification.

112: Why do you put all measured parameters in brackets? This is the most important information.

124-128: three times "was used" on 5 lines. Avoid repetitions.

Section 3.3.2 describes the decomposition of spring hydrographs after Jeannin & Sauter in a very general way, but it is not clear if and how this approach was used

in the present study. Similar problem in section 3.3.3. In the "material and methods" chapter, please always say clearly what you did in your study, and how you did it, instead of describing general theory.

Heading 4.1 could be shortened.

217: "between 44 and more than 50 Mm$^3$" (call me pedantic, but "between 44 and > 50 Mm$^3$" is an improper use of language and mathematical symbols).

290: "increases the reduction" – slightly confusing. Better say "leads to stronger reduction".

Discussion: Very good. Here, you discuss three main aspects of your research. However, you have only formulated two research objectives. Wouldn't it be better to have at least three major research objectives, corresponding to these three main aspects? See my comment above.

320-323: Very long and extremely intricate sentence that contains a surprisingly insignificant message. Please split into several sentences, rearrange and rephrase.

Conclusions: Already in the first sentence of the conclusions, you undersell the importance of your study with respect to climate change impacts on groundwater resources, because you only mention the sensitivity of your MODEL to climatic conditions, which is a rather academic perspective. However, climate change impacts on groundwater resources is a major topic, particularly in the Mediterranean area. I would suggest to emphasize more clearly that your model allows to better predict climate change impacts on groundwater resources, and explain why this is important and how your model could help to make better management decisions. This is a general recommendation, not only concerning the conclusions, but also title, abstract and introduction.

References: Complete, relevant and up-to-date reference list.

Figure 1: The graphical quality of this map should be improved. The hatching for geological units is distracting. I would suggest to use transparent colors instead, on

top of some more intense grey shading showing the topography of the area.

The tables are very small, but I hope that their size will be increased in the final paper.

---

## Author Comment (AC1) · 16 Apr 2020

Dear Prof. Goldscheider,

We would like to thank you for your review and positive feedback. As suggested, we will rephrase some sentences to clarify the text, avoid repetitions, and correct some mistakes. Hereafter we describe the main modifications that will be made to the document based on your comments (major comments addressed are labeled C# and replies are labeled A#). Minor comments are labeled MC# with the reply directly underneath it.

[Figure]

C1: Title: The title is too long (almost three lines). Please shorten to two lines. A1: We agree that the title needs to be shortened, and we are considering to change it as follows, to also emphasize on the climate change aspect of the work (as mentioned in following comments): "Calibration of a lumped karst system model and application to the Qachqouch karst spring (Lebanon) under climate change conditions"

C2: Abstract: Something is wrong with the first sentence, which is also too long. Please rephrase. A2: We agree that this sentence is too long and needs clarification. To reflect more the climate change aspect of the study as well, this first sentence will be rephrased into: "Flow in complex karst aquifers is challenging to conceptualize and to model, especially in poorly investigated areas, in semi-arid climates, and under changing climatic conditions. It is yet necessary for implementing long term sustainable water management practices."

C3: 62-65: The research objectives are relevant, but maybe you could add 1-2 additional objectives. Objective no. 2 is very general and could be complemented by a more specific research question, also including the practical relevance of your research, such as the expected climate change impacts and the implications for freshwater availability. Furthermore, objective no. 2 is not completely clear. What do you mean by "its sensitivity" – the sensitivity of the model or of the karst aquifer? A3: We thank you for this useful comment. To homogenize with the discussion section, and as suggested, the objectives of the study (lines 62-65) will be rephrased as follows: "The objectives of this work were 1) to acquire new knowledge of the hydrodynamic functioning of a complex karst aquifer derived from statistical and correlation time series analyses, 2) to illustrate how a semi-distributed lumped model can be calibrated on the basis of this knowledge, and 3) to assess the impact of climate change on the spring hydrodynamic to provide insight on fresh water availability. The approach is demonstrated on the Qachqouch karst spring in the region north of Beirut (Lebanon), a Mediterranean region governed by semi-arid conditions."

C4: Section 3.3.2 describes the decomposition of spring hydrographs after Jeannin &

Sauter in a very general way, but it is not clear if and how this approach was used in the present study. Similar problem in section 3.3.3. In the "material and methods" chapter, please always say clearly what you did in your study, and how you did it, instead of describing general theory. A4: We agree that information about the use of these methods was missing, and as suggested, a sentence will be added to clarify the use of time series decomposition at the end of the section 3.3.2: "This time series decomposition was applied to the spring flow rates (box 3, Fig. 2) to verify if spring flows could be relevantly linked to three conceptual reservoirs." As well, the use of correlation analysis in the study will be detailed at the end of the section 3.3.3: "Auto-correlation and cross-correlation analyses were performed on precipitation and on the Qachqouch flow rates time series to estimate the memory effect of the system and the delay between precipitation and the hydrological response of the karst system (box 3, Fig. 2). Auto-correlation and cross-correlation of simulated flow rates were compared to those of the input data, as an additional validation method (box 4, Fig. 2)."

C5: Conclusions: Already in the first sentence of the conclusions, you undersell the importance of your study with respect to climate change impacts on groundwater resources, because you only mention the sensitivity of your MODEL to climatic conditions, which is a rather academic perspective. However, climate change impacts on groundwater resources is a major topic, particularly in the Mediterranean area. I would suggest to emphasize more clearly that your model allows to better predict climate change impacts on groundwater resources, and explain why this is important and how your model could help to make better management decisions. This is a general recommendation, not only concerning the conclusions, but also title, abstract and introduction. A5: We thank the reviewer for this useful comment. To emphasize the climate change aspect of our work, beside the modifications of the title and the clarification of the work objectives (previous comments), the first paragraph of the conclusion (lines 377-381) will be changed to: "This work aimed at acquiring new knowledge of the hydrodynamic functioning of a complex karst aquifer derived from statistical and correlation time series analyses to optimize the calibration of a semi-distributed lumped

model. The model developed for the Qachqouch karst spring in Lebanon (semi-arid climate) was used to assess the impact of climate change on the spring hydrodynamic behavior to provide insight on fresh water availability under climatic constraints. Flow rates were analyzed statistically for a better conceptualization of the system, to allow the calibration of a semi-distributed linear reservoir model. The model was then used to reproduce current conditions and to analyze the impact of dryer and warmer possible future climate conditions on flow rates." We will also modify the lines 394-396 to underline the use of the main conclusions of the climate change analysis for fresh water management: "The climate change simulations brought new insights about possible future spring flow conditions, therefore allowing decision makers to develop more adapted scenarios for long-term fresh water management. The next steps of management plans should entail coupling of climate change scenarios at the catchment scale with land use change scenarios to improve overall future predictions and investigate solutions to alleviate the expected future depletion of semi-arid karst aquifer systems."

C6: Figure 1: The graphical quality of this map should be improved. The hatching for geological units is distracting. I would suggest to use transparent colors instead, on top of some more intense grey shading showing the topography of the area. A6: We agree that the graphical quality of the figure 1 needs to be improved. As suggested, transparent color for the geological formations will be used in Fig. 1 rather than hatching, making the new version of the figure as per attachment

MINOR CORRECTIONS: MC1: 16, 17 and elsewhere: m3 should be m3 This will be done as suggested.

MC2: 73: km2 should be km2 This will be done as suggested.

MC3: 80: Rearrange sentence to avoid misunderstanding. The spring is located at 64 m asl, not the aquifer. To avoid confusion about the spring elevation, the misleading sentence L80 will be rearranged into: "Similar to the nearby Jeita spring (Margane et al., 2013, 2018), the Qachqouch spring is located at 64 m asl and originates from the

Jurassic karst aquifer (Fig. 1)."

MC4: 84: quaternary should be Quaternary This will be done as suggested.

MC5: 84: The expression "high level of karstification" is misleading in this case. In fact, the Messinian salinity crisis created a very low topographic level of karstification. You probably mean high degree of karstification, very intense and very deep karstification. As suggested, L84 will be corrected to: "and Quaternary glaciations also contributed to creating a high degree of karstification (very intense and very deep) in the Mediterranean area in several stages"

MC6: 112: Why do you put all measured parameters in brackets? This is the most important information. As suggested, the list of measured parameters will be taken out of the brackets.

MC7: 124-128: three times "was used" on 5 lines. Avoid repetitions. To avoid repetitions, L124-128 will be rewritten into: "Spring flow rates were evaluated by a frequency analysis (Dörfliger et al., 2010; Mangin, 1971; Marsaud, 1997). Flow rates and their frequency of measurements were linked with a log normal distribution, except for outliers arising from variation in flow dynamics. Following hydrograph decomposition, the method developed by Mangin (1971, 1975) was used to estimate the dynamic volume (Vdyn) available in the aquifer during the depletion flow of a karst spring."

MC8: Heading 4.1 could be shortened. The heading 4.1 will be shortened to: "4.1 Qualitative assessment of the system"

MC9: 217: "between 44 and more than 50 Mm3" (call me pedantic, but "between 44 and > 50 Mm3" is an improper use of language and mathematical symbols). This will be done as suggested.

MC10: 290: "increases the reduction" – slightly confusing. Better say "leads to stronger reduction". This will be done as suggested.

MC11: 320-323: Very long and extremely intricate sentence that contains a surprisingly

insignificant message. Please split into several sentences, rearrange and rephrase. The sentence of L320-323 will be re-written into: "Even though the Qachqouch karst system has been reported to be less complex than that of the neighboring Jeita spring (Doummar, 2012; Margane et al., 2018), it is still comparable to other Middle Eastern karst systems (Fig. 5). Parameters k and i, representing the extent of the phreatic zone, the regulating capacity of the system, and the type of infiltration (Bakalowicz et al., 2008; El-Hakim and Bakalowicz, 2007; Mangin, 1975), classify the Qachqouch spring close to other Lebanese karstic aquifers."

[Figure]

[Figure]

**Fig. 1.** Location of the Qachqouch karst spring near Beirut (Lebanon).

---

## Referee Comment (RC2) · Anonymous Referee #2 · 16 Jun 2020

In the submitted manuscript, Dubois et al apply a karst simulation model at a large karst spring in the north of Beirut (Lebanon). The model is calibrated and evaluated with discharge observations at the spring supported by spring flow characterization methods. The model is then used to assess the impact of climate change of discharge using perturbations of the historic climate conditions that were derived from climate projections. A strong dependency of spring discharge on precipitation compared to a rather weak influence of increasing temperatures on spring discharge. The authors also find some indication for increased flood frequencies but acknowledge remaining

uncertainties in the simulated recharge vernation processes.

This study presents an elegant new approach to use methods, originally developed for system characterization, to improve the setup of a karst simulation model. The paper is written well and concise. The results are plausible and are discussed with respect to the work of others. For all those reasons, I think it will make a valuable contribution to Hydrology and Earth System Sciences. However, in order to reach publication quality, the paper needs some minor to moderate revisions as elaborated below and in the attached pdf:

- A lot of important information is provided in the introduction but a clear research gap still needs to be defined, which the authors intent to fill with this particular work.

- Some clarification of the spatial discretization of the model is necessary. MIKE-SHE is a distributed model, which is here applied fully distrusted at the surface but it is operating completely lumped in the subsurface of a set of sub-catchments. Using a sub-catchment approach, this seems to be a semi-distributed application of the MIKE-SHE model using a fully distributed surface routine, right?

- Mangin's method and the decomposition of spring hydrographs are usually applied to hydrograph recessions. Please elaborate how recessions were defined/extracted for the entire time series.

- Please explain in more detail the model calibration procedure and how it is linked to the spring flow characterization.

- Some clarification on how and how many scenarios were derived from the IPCC projections for the climate change analysis. Mentioning table 2 already here might be helpful. In many regions, climate change is projected to have strongly different effects on P and T throughout the seasons. Why did this study choose a delta approach for entire years?

- In the results/discussion, the link between model structure and spring flow character-

ization is not very clear.

- Can you provide a sketch of the conceptual model of the system?

Please also note the supplement to this comment:
https://www.hydrol-earth-syst-sci-discuss.net/hess-2020-90/hess-2020-90-RC2-supplement.pdf

―――――――――――――――――――

**Supplement:**

[revised manuscript text omitted]

---

## Author Comment (AC2) · 22 Jun 2020

Answer to comments of Anonymous reviewer 2 -2020-06-16

Dear Referee,

We would like to thank you for your comments and positive review that aimed at improving our paper. As suggested, we have reviewed the text and completed the missing information. Hereafter we describe the main modifications that will be made to the document based on your comments. The interactive comments are addressed
first (labeled C#), followed by comments from the annotated manuscript (labeled mC#, comments from the supplementary material).

C1: A lot of important information is provided in the introduction but a clear research gap still needs to be defined, which the authors intent to fill with this particular work. A1: We thank the Referee for this useful comment, and we will provide the missing information by adding L61: "Nevertheless, little information is found in the literature about the use of time series analysis in groundwater flow modeling for complex karst aquifers to enhance model calibration and estimate the sensitivity of water resources to climate change while lumped models seem to be a good tool to address this question."

C2: Some clarification of the spatial discretization of the model is necessary. MIKE-SHE is a distributed model, which is here applied fully distrusted at the surface but it is operating completely lumped in the subsurface of a set of sub-catchments. Using a sub-catchment approach, this seems to be a semi-distributed application of the MIKESHE model using a fully distributed surface routine, right? A2: That is exact, and we agree that clarification is needed on the spatial discretization of the model. Indeed, only the surface routine is fully distributed (atmosphere and unsaturated zone). Lumped reservoirs are used for the saturated zone (at the subcatchment scale – for the five interflowing reservoirs I1 to I5 in Fig.3) and for the entire system with two baseflow reservoirs for the entire study area (reservoirs B1 and B2 in Fig. 3). In order to clarify this point, Fig. 3 will be modified to add the spatialization details at each level, as follows.

Additionally, the L67 (Introduction) will be rephrased as follows: "A semi-distributed/lumped model, composed of a spatially distributed superficial and unsaturated zone, and a saturated zone composed of interflowing lumped reservoirs, developed using MIKE SHE (DHI, 2016a, b), is calibrated here using observed spring discharge time series."

C3: Mangin's method and the decomposition of spring hydrographs are usually applied

to hydrograph recessions. Please elaborate how recessions were defined/ extracted for the entire time series. A3: We agree that information on how recessions were estimated from the time series is missing. We will therefore add L142: "Considering the very similar annual recession for the spring and summer period in 2015, 2016, and 2017, annual recession was considered from the last major flood (occurring in March/April) until the first rainfall event in October (Fig. 2)."

C4: Please explain in more detail the model calibration procedure and how it is linked to the spring flow characterization. A4: Indeed, the explanation of how the model calibration is linked to the spring flow characterization was missing. In order to rectify this, L195 will be rewritten as follows: "The model uses 20 parameters defining the atmosphere, the unsaturated zone, the saturated interflow reservoir, and the base flow reservoirs. The time constant of each reservoir from the unsaturated zone is determined from the flow characterization and the recession analysis, therefore reducing the uncertainty in some parameters and the total number of parameters for calibration."

C5: Some clarification on how and how many scenarios were derived from the IPCC projections for the climate change analysis. Mentioning table 2 already here might be helpful. In many regions, climate change is projected to have strongly different effects on P and T throughout the seasons. Why did this study choose a delta approach for entire years? A5: We agree that a clarification on the climate scenarios is needed, and that a mention of Table 2 in section 3.3.5 would be helpful. For that purpose, we will add after L212: "A combination of those conditions for ten consecutive years (annual gradient of warming temperature and decreasing precipitation) were applied to the average year (derived from averaging the monitored daily precipitation and temperature) to obtain seven scenarios of changing climatic conditions for the 2030 horizon (Table 2). If annual gradients have been chosen for simplification purposes, the changing conditions in semi-arid conditions actually concern the rainy season (October to April of the following year) when exchanges between the atmosphere and the system are active (runoff, evapotranspiration, infiltration. . .)."

C6: In the results/discussion, the link between model structure and spring flow characterization is not very clear. A6: We will include this recommendation and rephrase L270-273 as follows: "The previous analyses performed on the time series allowed refining the model geometry by matching the number of reservoirs with the conclusions of the flow characterization. Model parameterization also included setting to fixed values the time constants of the reservoirs, which are usually included in the calibration process. Model uncertainty has thus been reduced by optimizing the conceptual model on the system hydrodynamic functioning (Enemark et al., 2019)."

C7: Can you provide a sketch of the conceptual model of the system? A7: We thank the Referee for their suggestion. However, we believe that a conceptual model of the system by itself would be very similar to Fig. 3 and we chose not to add a new figure. However, we will add more details to Fig. 3 to make it clearer and underline in the text that this figure represents a conceptual model of the system. Please refer to A2 for the modified figure and L333-342 will be rephrased: "Although the model adequately reproduces flow discharge, it underestimates the summer low flows. Measurements recorded during flooding of the spring gauging station might be underestimated due to errors in the discharge water level rating curve for high flow rates. Another explanation could be that the fast flow linked to a highly-developed drainage system is oversimplified in this reservoir model (reservoirs I1 to I5 from Fig. 3). The thickness of the UZ, combined with its lithological heterogeneity, as well simplified in the model (Fig. 3), may contribute to the relatively stable summer low flow by allowing considerable water storage, which is represented by the B2 baseflow reservoir in the model (Fig. 3). In fact, the dolostone could be compared to low-permeability porous media drained by a high-permeability system, thus allowing a large storage capacity in the upper parts of the aquifer. Furthermore, the high degree of karstification of the area, resulting from both the eustatic variations (Messinian) and the quaternary glaciations, leads to a complex drainage system, with three identifiable flow components (fast flow, intermediary flow, and baseflow – Fig. 3) and a probable paleo-network under the current base level. This would enhance the storage capacity of the system, as well as induce rapid flow

rate increases (Bakalowicz, 2015; Nehme et al., 2016)."

Revision from the additional material: mC1: L14 and 62: "a semi-distributed lumped model" – this is contradictory. Pick one: either semi-distributed or lumped. The text will be changed to "lumped model".

mC2: L21: "Climate change conditions (+1 to +3°C warming, -10 to -30% less precipitation annually, and intensification of rain events)" – in which future? 50 years? 100 years? The text will be rephrased into "Climate change conditions at the 2030 horizon (+1 to +3°C warming, -10 to -30% less precipitation annually, and intensification of rain events)"

mC3: L24: "with flow rates decreasing by 34%" – corresponding to what change of Precip? The text will be rephrased into "with flow rates decreasing by 34 % for scenarios with 30 % loss of yearly precipitation

mC5: p°2: A lot of important information is provided in the introduction but a clear research gap still needs to be defined, which the authors intent to fill with this particular work. Please refer to A1 for our answer.

mC6: L67: "A semi-distributed lumped model developed using MIKE SHE" – Isn't MIKE-SHE a distributed model? I think you need to define early in the manuscript what type of model you use. Please refer to A2 for our answer.

mC7: L127: "Following hydrograph decomposition, the method developed by Mangin (1971, 1975)" – Mangin's method is applied to hydrograph recessions. Please elaborate how recessions were defined for the entire time series. Please refer to A3 for our answer

mC8: section 3.3.2: Same as above [mC7]: How were recessions extracted? To define how recessions were extracted, we will add this text after L160: "To decompose the discharge flow from the spring accordingly to this concept, all decreasing parts of the discharge flow between 2015 and 2018 were used. The peaks higher than 10 m3/s

were excluded because of the incertitude on the flow measurement, as well as the portions of sub-vertical and too irregular decreasing slope."

mC9: L180: "into several sub-catchments" – How many? The text will be changed for: "into five sub-catchments"

mC10: L194: "The complete model is therefore considered a classical lumped model for the saturated zone" – but only within each sub-catchment, right? Please refer to A2 for our answer

mC11: L195: "physically-based model" – better use the term "fully distributed" here because it's about how the model is discretized for its different domains. The text will be rephrased for "fully distributed".

mC12: L199-200: "Sensitivity analysis was conducted automatically on single parameters using the Autocal function (DHI, 2016) to identify the parameters to which the model is highly sensitive." – Please add some more detail on how this sensitivity analysis works? Following L200, this text will be added: "The Autocal function performs a local sensitivity analysis by computing the ratio of the perturbation in the simulated discharge flow with the variation of a single parameter, one at a time."

mC13: section 3.3.5 – Please clarify in a bit more detail on how and how many scenarios were derived from the IPCC projections. A table might be helpful here. Please refer to A5 for our answer

mC14: section 4.1.1 – Please explain this method a bit better in the methodology [Flow rate frequency]. We will add the following text after L126: "The evolution of the slope of the curve between the breaking points gives information about the dynamic of the system and the time series. Dörfliger et al. (2010) classified the possible configuration and their respective interpretation."

mC15: L275-281 – Please move interpretations to discussion. The aforementioned paragraph only presents the comparison of the auto-correlation and cross-correlation

of the simulated and observed spring flow, therefore meaning that the model reproduces the spring discharge correctly. We do not think that this paragraph would fit in section 5 since it does not bring a global and interpretative point of view on the functioning of the system. Therefore, we prefer to keep it as it is.

mC16: L283: "from seven potential scenarios for the study area (Table 2)" – these have to be introduced and elaborated in detail in the methods section. In many regions, climate change is projected to have strongly different effects on P and T throughout the seasons. Why did this study choose a delta approach for entire years? Please refer to A5 for our answer.

mC17: section 5.2 – Can you provide a sketch of that? Please refer to A7 for our answer.
* * *
[Figure]

**Rainfall / snow**

**Evapotranspiration /
Vegetation interception /
Runoff**

**Topography**

**Atmosphere
(spatialized)**
Spatial resolution
50x50 m

**Subcatchments**

1 2 3 4 5

**Unsaturated zone
(spatialized)**
Spatial resolution
50x50 m

I5

**Saturated zone
(reservoirs – semi-
distributed)**

I4

**Interflow
reservoirs
(1/subcatchment)**

I3

Fast flow – leading to
Peak flow rates

I2

I1

B1

**Baseflow
reservoirs**

Intermediary flow –
intermediate
depletion flow

**Qachqouch
spring**

B2

Baseflow–
Continuous spring
base flow

**Fig. 1.**